# Synthesis and NLRP3-Inflammasome Inhibitory Activity of the Naturally Occurring Velutone F and of Its Non-Natural Regioisomeric Chalconoids

**DOI:** 10.3390/ijms23168957

**Published:** 2022-08-11

**Authors:** Tiziano De Ventura, Mariasole Perrone, Sonia Missiroli, Paolo Pinton, Paolo Marchetti, Giovanni Strazzabosco, Giulia Turrin, Davide Illuminati, Virginia Cristofori, Anna Fantinati, Martina Fabbri, Carlotta Giorgi, Claudio Trapella, Vinicio Zanirato

**Affiliations:** 1Department of Chemistry, Pharmaceutical and Agricultural Sciences, University of Ferrara, Via Luigi Borsari 46, 44121 Ferrara, Italy; 2Department of Medical Sciences, Section of Experimental Medicine, University of Ferrara, Via Fossato di Mortara, 64/b, 44121 Ferrara, Italy; 3Laboratory for Technologies of Advanced Therapies (LTTA), Via Fossato di Mortara, 70, 44121 Ferrara, Italy; 4Maria Cecilia Hospital, GVM Care & Research, 48033 Cotignola, Italy; 5Department of Environmental and Prevention Sciences, University of Ferrara, Via Fossato di Mortara 17, 44121 Ferrara, Italy

**Keywords:** flavonoid, chalcone-based compounds, NLRP3-inflammasome inhibitors

## Abstract

Plant-derived remedies rich in chalcone-based compounds have been known for centuries in the treatment of specific diseases, and nowadays, the fascinating chalcone framework is considered a useful and, above all, abundant natural chemotype. Velutone F, a new chalconoid from *Millettia velutina*, exhibits a potent effect as an NLRP3-inflammasome inhibitor; the search for new natural/non-natural lead compounds as NLRP3 inhibitors is a current topical subject in medicinal chemistry. The details of our work toward the synthesis of velutone F and the unknown non-natural regioisomers are herein reported. We used different synthetic strategies both for the construction of the distinctive benzofuran nucleus (**BF**) and for the key phenylpropenone system (**PhP**). Importantly, we have disclosed a facile entry to the velutone F via synthetic routes that can also be useful for preparing non-natural analogs, a prerequisite for extensive SAR studies on the new flavonoid class of NLRP3-inhibitors.

## 1. Introduction

Chalcone-based compounds can act as photoinitiators of polymerization under visible light with an excellent profile for (i) free radical polymerization, (ii) cationic polymerization, (iii) synthesis of interpenetrating polymer networks (IPNs), and (iv) thiol-ene reactions [1]. Additionally, fluorescent chalcone derivatives have been used for the development of a mouse embryonic stem cell probe [2]. On the other hand, in-depth pharmacological studies concluded that natural extracts containing chalcone-based compounds exhibit an impressive array of biological activities, including anti-inflammatory/anticancer effects [3,4,5]. Actually, chalcones may inhibit specific enzymes such as different kinases [6,7,8], the aldose reductase [9,10], cyclooxygenases [11,12], and inducible nitric oxide synthase [13,14]. Moreover, it has been shown that both synthetic and natural chalconoids play a healthy role in several diseases by inhibiting the NLRP3-inflammasome formation [15,16,17,18,19]. The NLRP3 inflammasome is a large protein complex controlling the production of caspase-1 and ultimately of pro-inflammatory cytokines (IL-1 and IL-18). In this context, it was reported that velutone F (**1**) (Figure 1), a retrochalcone [20,21,22] recently identified in the ethanolic extract of the leguminous plant *Millettia velutina* [23,24], inhibits the formation of the NLRP3 active complex. Among the eight new flavonoids identified in the lipophilic crude residue derived from 10 kg of dry vine stems of *Millettia velutina*, compound **1** exhibited the most potent inhibitory effect against nigericin-induced IL-1 release in THP-1 cells. Velutone F, featuring the 1-phenyl-2-propen-1-one moiety (**PhP**) and a substituted benzofuran core (**BF**), can be classified as a hybrid chalcone. The development of hybrid molecules incorporating different pharmacophores, each with its own molecular target, is an important area of research in medicinal chemistry [25]. Actually, both natural and synthetic benzofuran-derived compounds have potential therapeutic interests ranging from antibacterial, antifungal, anti-inflammatory, analgesic, antidepressant, anticonvulsant, anticancer, anti-HIV, antidiabetic, antituberculosis, and antioxidant [26,27,28,29].

We are currently involved in a multidisciplinary study aimed at identifying new anti-inflammatory/anticancer compounds that mimic the MCC950 molecular structure (Figure 1). It has been demonstrated that the diarylsulfonylurea MCC950 powerfully inhibits the NLRP3 activation selectively [30,31]. In detail, MCC950 would seem to reversibly bind the NLRP3 multi-protein complex making it unable to generate the active complex, namely the NLRP3-inflammasome [32,33]. Because of the profoundly different chemical structures of the synthetic MCC950 compared to the one of velutone F (**1**), it is reasonable to assume that they can block the activation of the NLRP3-inflammasome by interfering with different bio-chemical targets. We were intrigued by the possibility of disposing of multi-milligrams amount of the natural substance **1** for the purpose of undertaking pharmacochemical investigations and mostly to shed some light on the mechanism by which compound **1** inhibits the nigericin-induced IL-1 release. Actually, when we started the search, the chemical identity of **1** was ascertained by spectroscopic means exclusively, primarily 2D NMR [23]. However, very recently [24], the same research teams got proof of the chalconoid structure of compound **1** by its semi-synthesis from Khellin.

The reasons mentioned above prompted us to design/develop synthetic routes for preparing both the natural substance originally extracted from the tropical plant *Millettia velutina* and analogs to be studied by the biologist team as promising anti-inflammatory agents.

## 2. Results and Discussion

A host of synthetic strategies has been developed to create the trans-carbon-carbon double bond of the 1,3-diaryl-2-propen-1-one moiety featuring chalcone compounds, including Claisen-Schmidt’s condensations, Wittig, and Julia-Kocienski olefinations. Instead, palladium-catalyzed reactions, namely Heck, Sonogashira, and Suzuki-Miyaura cross couplings, have been used to establish the 1,3-diaryl-enone framework through A-ring/C-1 and/or C1/C2 bond formation [34].

Initially, we planned a synthetic strategy to compound **1** based on the introduction of the required alkene by Wittig olefination of 5-formyl-4,7-dimethoxy benzofuran **7** (**5-FBF**) with the 1-phenyl-2-(triphenylphosphoranylidene)ethanone counterpart **8** [35]. To this end we elaborated two synthetic approaches for the preparation of the key intermediate **5-FBF** differing in the way the benzofuran core (**BF**) could be formed via annellation of the carbocyclic ring system onto a preformed furan, synthetic pathway A (Figure 1), or alternatively, by creating the furan ring by intramolecular cyclization on a preexistent carbocyclic, synthetic pathway B (Figure 2).

### 2.1. Synthetic Pathway A for Target Compound ***1***

The first route starts by treating the 3-furoic acid with excess LDA to produce the corresponding C-2 lithiated carboxylate lithium salt, which, by reacting with succinic anhydride, gave the dicarboxylic acid **2**. The desired acyl derivative **3** could be isolated in a modest yield after esterification of **2** [36,37]. The subsequent carbonyl group protection as dimethyl ketal **4** opened the way to the creation of the annellated six-membered carbocyclic ring. Thus, the Dieckmann cyclization, performed with potassium tert-butoxide at −78 °C, occurred with simultaneous elimination of MeOH from the dimethyl ketal group yielding the aromatic derivative **5**, which was taken to the benzofuran derivative **6** by etherification. Subsequently, the methoxycarbonyl group was converted to the required formyl group by a two-step process entailing reduction with LiAlH_4_ and oxidation of the resulting primary alcohol with pyridinium chlorochromate (PCC). The resulting **5-FBF** was eventually reacted with the stabilized phosphorous ylide **8**, in turn, prepared according to the literature [35]. The Wittig olefination under microwave irradiation provided velutone F (**1**) in 70% yield after chromatographic purification. As expected, NMR spectroscopic data for the synthesized velutone F were superimposable to the ones originally reported for the retrochalcone isolated from *Millettia velutina*.

### 2.2. Synthetic Pathway B for Target Compound ***1***

The starting move of the alternative synthetic approach to **7** entailing the creation of the furan ring by intramolecular cyclization of the bromobenzene derivative **11** (Figure 2) was the Dakin-like oxidation of the cheap 2,5-dimethoxybenzaldehyde by using the H_2_O_2_/cat. SeO_2_ system in *tert*-BuOH [38]. Methanolysis of the resulting arylformate promptly furnished the 2,5-dimethoxyphenol **9**, which underwent regioselective bromination with NBS affording **10**. The exclusive C-4 bromination accounted for the marked para orienting effect of the phenolic hydroxyl group [39].

At this stage, with the aim of creating the annellated 2,3-unsubstituted furan ring, we needed to introduce an O-tethered functionalized two carbon fragment. Thus, compound **10** was easily converted to the aryl ether **11** by treatment with bromoacetaldehyde dimethyl acetal and KOH in dimethylacetamide (DMA) [40]. The anticipated intramolecular electrophilic aromatic substitution was carried out with polyphosphoric acid (PPA), providing the benzofuran derivative **12** in a 50% yield. Transformation of the 5-bromo benzofuran derivative **12** into 5-formyl-4,7-dimethoxy benzofuran **7** (**5-FBF**) was achieved through halogen-metal exchange with BuLi followed by reaction with DMF [41].

In order to make the critical furan ring annellation step more efficient, we turned our attention to a recently reported protocol for the synthesis of 6-hydroxybenzofuran based on an unusual cycloaddition-cycloreversion sequence [42]. To this end, we prepared the penta-substituted benzene derivative **16** starting from the tri-substituted phenol derivative **9** (Figure 2), exploiting the well-known ability of the OTHP as an ortho-directing group for the metalation [43]. Accordingly, compound **9** was promptly transformed into the corresponding tetrahydropyranyl ether **14** that was at first ortho-lithiated by treatment with BuLi and later reacted with DMF to give the tetra-substituted phenyl derivative **15**. The subsequent bromination para to the phenol group [44] afforded the desired penta-substituted benzene **16**, to which the O-ethoxycarbonylmethylene fragment was easily inserted by standard etherification reaction. In previous saponification, compound **17** was treated with the Ac_2_O-AcONa system with heating to give the required 5-bromo benzofuran derivative **12** in an appreciable 57% yield. The mechanism proposed for the interesting cyclization reaction entails dehydration of the carboxyl group to give an unstable ketene intermediate that is trapped intramolecularly by the formyl group. The thermal [2 + 2] heterocycloaddition reaction is followed by a cycloreversion with the expulsion of CO_2_ and production of the 2,3-unsubstituted benzofuran derivative **12** [42].

Having the suitably derivatized aryl ring-B moiety in hand, we conceived preparing compound **1** by exploiting the Mizoroki-Heck cross-coupling reaction between **12** and 1-phenyl-2-propen-1-one **13**. The latter reagent was, in turn, easily prepared by reacting phosphorous ylide **8** and formaldehyde according to a known Wittig protocol [45] (Figure 2). Disappointingly, the Pd(0)-catalyzed reaction provided the retrochalcone **1** with a modest 32% yield [46].

### 2.3. Synthetic Pathways Providing the Non-Natural Regioisomers of Velutone F

With the aim of learning about stereo-electronic properties of the hybrid benzofuran-retrochalcone scaffold, we decided to prepare the non-natural compounds **22**, **23,** and **28** featuring the **PhP** moiety attached, respectively, at C-2, C-6, and C-3 of the 4,7-dimethoxy benzofuran core (**BF**). The non-natural regioisomers of velutone F are previously unknown compounds.

#### 2.3.1. Synthetic Pathways to the Isomers 22 and 23

The direct formylation of electron-rich arenes can be conveniently accomplished via the Vilsmeier–Haack (V–H) reaction. Indeed, the benzo[b]furan nucleus is reported to yield the 2-formyl derivative by reaction with the V–H electrophilic species [47]. We anticipated that the regioselectivity of the V–H reaction could change if electron-donating groups were present on the phenyl ring of the benzo-fused system. In line with our hypothesis, we decided exploring the behavior of 4,7-dimethoxy benzo[b]furan **19** under V-H reaction conditions (Figure 3). We planned to build the substituted benzo[b]furan **19** from **9** by creating the annellated 2,3-unsubstituted furan ring according to our previously sound synthetic pathway B for target compound **1** (Figure 2). Thus, once etherified the phenolic group of **9** with the functionalized two carbon fragment, the resulting compound **18** was cyclized to **19** under the action of PPA (Sn-β zeolite also showed to efficiently promote this transformation [48]). As expected, we found the subsequent electrophilic aromatic substitution reaction was poorly regioselective: all but one of the regioisomeric formyl benzofuran derivatives **2-FBF**, **5-FBF**, and **6-FBF** were formed. In detail, chromatographic purification of the residue from the V-H reaction led us to isolate compounds **20** (**2-FBF**) together with **7** (**5-FBF**) in 37% yield (^1^H NMR and HPLC analysis showed the isomers were in a 3.5:6.5 ratio), and compound **21** (**6-FBF**) in 30% yield. At this stage, we submitted the separated fractions to the Wittig olefination with the stabilized phosphorous ylide **8**. We obtained chalcone **23** from **6-FBF**, while in the same manner, the inseparable mixture of **2-FBF** and **5-FBF** furnished chalcones **22** and **1**, which, gratifyingly, could be easily separated by column chromatography.

#### 2.3.2. Synthetic Pathway to the Isomer **28**

We envisaged setting up the **PhP** moiety of **28** by reaction of 3-formyl benzofuran derivative **27** (**3-FBF**) with the ylide **8** according to the previously tested protocol entailing a classical Wittig olefination. About **3-FBF** preparation, we selected compound **25** as the direct precursor having in mind a controlled oxidation of its C-3 methyl substituent in order to derive the pivotal electrophilic functional group [49,50] (Figure 4). Thus, we devised preparing compound **25** in two steps from phenol **9**, namely: etherification with bromoacetone followed by acid-promoted cyclization of the resulting aryloxy acetone derivative **24**. The planned SeO_2_-oxidation of the C-3 methyl residue of compound **25** furnished a 1:1 mixture of the primary alcohol **26** and the aldehyde **27**, which were separable by chromatography. However, we found it very easy to carry out the oxidation of **26** to **3-FBF** by using the Corey–Suggs reagent (PCC). Eventually, the microwave-promoted Wittig reaction between the aryl aldehyde **27** (**3-FBF**) and the phosphorous ylide **8** provided the aimed chalconoid **28** in good yield.

#### 2.3.3. Inhibition of IL-1β Release In Vitro and In Vivo

Among the inflammasomes, NLRP3 is the most studied and characterized due to its implication in the pathogenesis of different human diseases [51]. The activation of NLRP3 inflammasomes consists of caspase-1 activation, which in turn induces secretion of the inflammatory cytokine IL-1β. Hence, IL-1β release is the most used read-out for NLRP3 inflammasome activation. In this study, IL-1β release was determined by ELISA assay to assess the inhibitory effects of synthesized compounds on NLRP3 inflammasome activation both in mouse bone marrow-derived macrophages (BMDMs) and in human PMA-differentiated lipopolysaccharide (LPS)-primed THP-1 macrophages (Figure 2). Velutone F (**1**) and the non-natural compounds **22**, **23**, and **28** demonstrated a high inhibitory capacity on the release of IL-1β.

To confirm the anti-inflammatory activity of these compounds in vivo, we used an LPS-induced inflammation treatment. Mice were intraperitoneally (IP) pre-injected with velutone F (**1**) and the non-natural isomers **22**, **23**, and **28** or vehicle at 25 mg/kg and then were IP injected with LPS (1 mg/kg). After 4 h, plasma and peritoneal exudate were collected from mice and analyzed for evaluation of IL-1β release. The mice that received pre-treatment with our compounds displayed a dramatic reduction in IL-1 production both in peritoneal exudate and blood samples (Figure 3).

Taken together, these results revealed that synthesized velutone F (**1**), as well as its regioisomers 22, 23, and 28, exerted a strong inhibition on the NLRP3 inflammasome activation both in vitro and in vivo.

## 3. Materials and Methods

### 3.1. Chemistry: Materials and General

All reagents and solvents that were commercially purchased were directly used without prior treatment. Reaction temperatures were recorded using a regular thermometer without correction. Melting points were determined on the Reichert Termovar apparatus and are uncorrected. Reactions were monitored by analytical thin-layer chromatography (TLC) on silica gel F254 glass plates (Merck, Darmstad, Germany) and visualized under UV light 254 nm and KMnO_4_. Mobile phases abbreviations: A = Ethyl acetate, P = petroleum ether, DCM = dichloromethane. ^1^H NMR spectra were recorded with a Mercury Place Varian 400 MHz NMR spectrometer (Varian Palo Alto, CA, USA) at room temperature. Chemical shifts (in ppm) were recorded as parts per million (ppm) downfield to tetramethylsilane (TMS). The following abbreviations are used for a multiplicity of NMR signals: s, singlet; d, doublet; t, triplet; q, quartet; m, multiplet; dd, double doublet; dt, double triplet; ddd, doublet of doublet of doublets; dtd, doublet of triplet of doublets. MS was carried out using electrospray ionization (ESI MICROMASS ZQ 2000, Waters, Milford, MA, USA). HRMS was acquired by CIGS University of Modena e Reggio Emilia (Modena, Italy) using a Q-Exactive Hybrid Quadrupole Orbitrap (Thermo Scientific, Waltham, MA, USA). IR spectra were taken on a Perkin-Elmer FT-IR spectrum 100 spectrometer (Foster City, CA, USA). Analytical HPLC was performed with Beckman System Gold 168 (Milan, Italy) using a C18 100 A Phenomenex Kinetex (150 × 4.6 mm) and UV-DAD detection. [A]: 0.1% TFA in water and [B]: 0.1% TFA in acetonitrile was used as a binary mobile phase at a flow rate of 0.7 mL/min. Gradient: from 0% to 100% [B] in 25 min. Microwave reactions were performed in a Biotage Initiator Plus reactor (Biotage, Sweden) using 2–5 mL glass vials with a rubber cap. For all experimental spectra see Appendix A.

#### 3.1.1. Methyl 3-(4-Methoxy-4-oxobutanoyl)furan-2-carboxylate (**3**) 

A solution of 3-Furoic acid (1 g, 8.92 mmol, 1 equiv.) in anhydrous THF (6 mL) was added dropwise, under argon atmosphere, to a cooled (−78 °C) solution of LDA, which was previously prepared from *n*-BuLi (12.5 mL, 1.6 M in hexane, 19.96 mmol, 2.2 equiv.) and *i*-Pr_2_NH (2.9 mL, 19.96 mmol, 2.2 equiv.) in THF (25 mL). The reaction mixture was stirred at −78 °C for 2 h, then a solution of succinic anhydride (1 g, 9.99 mmol, 1.1 equiv.) in anhydrous THF (8 mL) was added to the reaction flask. The cooling bath was removed, and aqueous 2 M HCl (30 mL) was added at room temperature. The mixture was extracted with chloroform, the organic phase was dried over Na_2_SO_4,_ and solvent was evaporated. The solid residue was dissolved in MeOH (50 mL), cooled at 0 °C, and H_2_SO_4_ (0.45 mL, 95%) was added dropwise. The mixture was stirred at room temperature for 24 h, then a saturated NaHCO_3_ solution was added, and the mixture was extracted with AcOEt. The organic phase, dried over Na_2_SO_4_, was evaporated under reduced pressure to give a crude that was purified by silica gel column chromatography with A/P 1:9, *v*/*v* as a solvent system. Title compound (**3**) was isolated as a colorless oil. Yield 32% over two steps. ^1^H NMR (400 MHz, Chloroform-*d*) δ 7.51 (d, *J* = 1.7 Hz, 1H), 6.81 (d, *J* = 1.7 Hz, 1H), 3.89 (s, 3H), 3.68 (s, 3H), 3.31 (t, *J* = 6.7 Hz, 2H), 2.73 (t, *J* = 6.6 Hz, 2H). ^13^C NMR (101 MHz, Chloroform-*d*) δ 187.72, 172.96, 162.89, 150.83, 144.36, 122.29, 113.62, 52.46, 51.83, 34.92, 27.47 [36,37]. 

#### 3.1.2. Methyl 3-(1,1,4-Trimethoxy-4-oxobutyl)furan-2-carboxylate (**4**) 

A solution of compound **3** (0.9 gr, 3.75 mmol, 1 equiv.), HC(OMe)_3_ (2.1 mL, 19.48 mmol, 5.2 equiv.), and *p*-TsOH (0.05 gr) in MeOH (8 mL) was refluxed for 48 h. The reaction mixture was cooled to room temperature, pyridine (0.18 mL) was added, and the solvent evaporated.The residue was purified by silica gel column chromatography with A/P 1:4, *v*/*v* as a solvent system. Title compound **4** was obtained as a transparent oil. Yield 90%. ^1^H NMR (400 MHz, Chloroform-*d*) δ 7.36 (d, *J* = 1.7 Hz, 1H), 6.66 (d, *J* = 1.7 Hz, 1H), 3.81 (s, 3H), 3.58 (s, 3H), 3.20 (s, 6H), 2.47 (t, *J* = 6.7 Hz, 2H), 2.16 (t, *J* = 6.6 Hz, 2H). ^13^C NMR (101 MHz, Chloroform-*d*) δ 173.51, 163.81, 154.87, 141.84, 117.12, 112.38, 111.59, 101.72, 52.18, 51.97, 49.51, 30.35, 28.92 [36,37].

#### 3.1.3. Methyl 4-Hydroxy-7-methoxybenzofuran-5-carboxylate (**5**)

*tert*-BuOK (0.61 g, 5.45 mmol, 2.2 equiv.) was poured into anhydrous THF (65 mL) under an argon atmosphere, and the suspension was cooled to −78 °C. A solution of compound **4** (0.71 g, 2.48 mmol, 1 equiv.) in anhydrous THF (2 mL) was added dropwise, and the mixture turned orange. After 1.5 h of stirring at −78 °C, anhydrous HCl (4 M in 1,4-dioxane) was added. The resulting transparent yellow reaction mixture was allowed to room temperature and stirred for 1 h. The precipitate was filtered through a Gooch filter, and the solvent was evaporated under reduced pressure. Chromatographic purification of the residue with A/P 1:9 *v*/*v* as the eluent phase provided title compound **5** as a white solid. Melting point: 125–127 °C. Yield 62%. ^1^H NMR (400 MHz, Chloroform-*d*) δ 11.14 (s, 1H), 7.60 (dd, *J* = 2.1 Hz, 1H), 7.17 (s, 1H), 6.98 (d, *J* = 2.1 Hz, 1H), 3.97 (d, *J* = 1.6 Hz, 6H). ^13^C NMR (101 MHz, Chloroform-*d*) δ 171.09, 152.43, 149.52, 144.76, 139.19, 118.90, 105.66, 105.38, 105.02, 56.51, 52.29 [36,37].

#### 3.1.4. Methyl 4,7-Dimethoxybenzofuran-5-carboxylate (**6**) 

A mixture of compound **5** (0.21 g, 0.95 mmol, 1 equiv.), Cs_2_CO_3_ (1.55 g, 5.75 mmol, 5 equiv.), and CH_3_I (0.29 mL, 4.75 mmol, 5 equiv.) in anhydrous DMF (6 mL) was stirred at room temperature for 16 h under an argon atmosphere. After complete conversion of starting material (checked by TLC A/P9 1:9), the reaction was partitioned between HCl 2 M and AcOEt; the organic phase was washed with HCl 2 M (×2) and brine, dried over Na_2_SO_4,_ and the solvent was evaporated to dryness. Purification of the crude by column chromatography with A/P 1:4 as the eluent phase afforded title compound **6** as a colorless oil. Yield 96%. ^1^H NMR (300 MHz, Chloroform-*d*) δ 7.63–7.61 (m, 1H), 7.26 (s, 1H), 6.95–6.94 (m, 1H), 4.00 (d, *J* = 0.8 Hz, 6H), 3.94 (d, *J* = 0.8 Hz, 3H). ^13^C NMR (75 MHz, Chloroform-*d*) δ 167.04, 149.35, 148.04, 145.57, 141.84, 123.31, 117.58, 108.69, 105.80, 62.46, 56.77, 52.52.

#### 3.1.5. 4,7-Dimethoxybenzofuran-5-carbaldehyde (**7**)

A solution of compound **6** (0.2 g, 0.84 mmol, 1 equiv.) in dry Et_2_O (4 mL) was added dropwise to a stirred to a cooled (0 °C) suspension of LiAlH_4_ (0.08 g, 2.1 mmol, 2.5 equiv.) in anhydrous Et_2_O (5 mL) under argon atmosphere. The reaction mixture was stirred at room temperature overnight, then it was cooled to 0 °C, and aqueous NaOH solution (5 mL, 2 M) was added. After 30 min, the mixture was filtered on a celite pad, and the phases were separated. The aqueous phase was extracted with Et_2_O. The combined organics were dried over Na_2_SO_4_, and the solvent was evaporated to give (4,7-dimethoxybenzofuran-5-yl)methanol as a white crystalline solid, which was used in the next oxidative step without further purification.

A solution of the primary alcohol (0.14 g, 1.29 mmol, 1 equiv.) in anhydrous DCM (5 mL) was added to a suspension of PCC (0.42 g, 1.93 mmol, 1.5 equiv.) in DCM (5 mL). The dark brown reaction mixture was magnetically stirred at room temperature for 1.5 h then Et_2_O was added. The supernatant was decanted from the black insoluble residue, which was further washed with Et_2_O (×3). The combined organic phases were filtered through a pad of silica, and the solvent was evaporated. Purification of the crude by column chromatography with A/P 1:4 *v*/*v* as a solvent system furnished title compound **7** as a white crystalline solid. Melting point: 142–145 °C. Yield 50%.^1^H NMR (400 MHz, Chloroform-*d*) δ 10.46 (s, 1H), 7.67 (dd, *J* = 2.3, 0.4 Hz, 1H), 7.25 (s, 1H), 7.02 (d, *J* = 2.3 Hz, 1H), 4.15 (s, 3H), 4.00 (s, 3H). ^13^C NMR (101 MHz, Chloroform-*d*) δ 189.09, 149.92, 145.30, 142.04, 122.24, 120.30, 105.88, 103.45, 61.74, 56.29.

#### 3.1.6. 2,5-Dimethoxyphenol (**9**) 

H_2_O_2_ (2.4 mL, 30% *w*/*w*, 78.21 mmol, 5.4 equiv.) was added dropwise to a magnetically stirred mixture of 2,5-dimethoxybenzaldehyde (2.42 g, 14.56 mmol, 1 equiv.), SeO_2_ (0.038 g, 0.342 mmol, 0.023 equiv.) in *tert*-BuOH (6 mL). The reaction mixture was stirred overnight at room temperature then Et_2_O (20 mL) was added. The resulting mixture was washed with brine (×3), the organic phase was dried over Na_2_SO_4_, and the solvent was evaporated to dryness. The residue was dissolved in MeOH (10 mL), and a solution of K_2_CO_3_ (2 g) in H_2_O (10 mL) was added to the solution. After 1 h, HCl 6M was added to reach pH 1, and the reaction mixture was extracted with DMC (20 mL ×3). The combined organic phases were dried over anhydrous Na_2_SO_4,_ and the solvent was evaporated to dryness. Purification was carried out by silica gel column chromatography with A/P 1.5:8.5 *v*/*v* to obtain compound **9** as a transparent oil. Yield 90%. ^1^H NMR (400 MHz, Chloroform-*d*) δ 6.77 (d, *J* = 8.8 Hz, 1H), 6.56 (d, *J* = 2.9 Hz, 1H), 6.38 (dd, *J* = 8.8, 2.9 Hz, 1H), 5.66 (s, 1H), 3.84 (s, 3H), 3.75 (s, 3H). ^13^C NMR (101 MHz, Chloroform-*d*) δ 154.56, 146.44, 140.96, 111.45, 104.23, 101.73, 56.58, 55.66 [38].

#### 3.1.7. 4-Bromo-2,5-dimethoxyphenol (**10**)

NBS (1.16 g, 6.55 mmol, 1 equiv.) was added to a cooled (0 °C) solution of **9** (1.0 g, 6.55 mmol, 1 equiv.) in acetonitrile (130 mL). Stirring at 0 °C was continued for 30 min. Then, Na_2_S_2_O_3_ saturated solution was added, and the mixture was extracted with AcOEt (×3). The combined organic phases were washed with brine, dried over Na_2_SO_4_, and the solvent was evaporated. Purification of the residue by column chromatography by eluting with A/P 1:4 *v*/*v* gave **10** as a transparent oil. Yield 97%. ^1^H NMR (400 MHz, Chloroform-*d*) δ 7.02 (s, 1H), 6.61 (s, 1H), 5.65 (s, 1H), 3.83 (s, 3H), 3.81 (s, 3H). ^13^C NMR (101 MHz, Chloroform-*d*) δ 150.66, 145.71, 140.94, 115.77, 100.51, 99.81, 56.77, 56.74.

#### 3.1.8. 1-Bromo-4-(2,2-dimethoxyethoxy)-2,5-dimethoxybenzene (**11**)

BrCH_2_CH(OMe)_2_ (1.0 mL, 8.82 mmol, 1.5 equiv.) was added dropwise to a stirred solution of **10** (1.37 g, 5.88 mmol, 1 equiv.) and KOH (0.66 g, 11.756 mmol, 2 equiv.) in DMAC (12 mL). The reaction mixture was stirred at 140 °C for 1.5 h and then cooled to room temperature. H_2_O (20 mL) was added, and the resulting mixture was extracted with Et_2_O (40 mL ×3). The combined organic phases were washed with aqueous NaOH (30 mL, 5%) and H_2_O (30 mL), dried over Na_2_SO_4,_ and the solvent was evaporated to dryness. Purification of the residue by column chromatography with A/P 1:4 *v*/*v* as the eluent furnished **11** as a white crystalline solid. Melting point: 57–60 °C. Yield 83%. ^1^H NMR (400 MHz, Chloroform-*d*) δ 7.05 (s, 1H), 6.63 (s, 1H), 4.72 (t, *J* = 5.2 Hz, 1H), 4.04 (d, *J* = 5.2 Hz, 0H), 3.83 (s, 3H), 3.80 (s, 3H), 3.46 (s, 6H). ^13^C NMR (101 MHz, Chloroform-*d*) δ 150.30, 148.22, 144.66, 117.47, 102.52, 102.07, 69.94, 57.10, 56.87, 54.51.

#### 3.1.9. 5-Bromo-4,7-dimethoxybenzofuran (**12**) by Cyclization of (**11**)

A solution of compound **11** (0.37 g, 1.55 mmol, 1 equiv.) in chlorobenzene (3 mL) was added dropwise to a heated (120 °C) mixture of polyphosphoric acid (0.5 g) in chlorobenzene (15 mL). After 30 min, the reaction mixture was cooled to room temperature, diluted with Et_2_O, and washed with H_2_O. The removal of volatiles under reduced pressure gave a residue that was purified by column chromatography with A/P 0.5:9.5 *v*/*v*. Compound **12** was isolated as a white crystalline solid. Melting point: 63–65 °C. Yield 50%. ^1^H NMR (400 MHz, Chloroform-*d*) δ 7.59 (dd, *J* = 2.2, 0.5 Hz, 1H), 6.92 (s, 1H), 6.87 (dd, *J* = 2.1, 0.4 Hz, 1H), 3.97 (d, *J* = 1.7 Hz, 6H). ^13^C NMR (101 MHz, Chloroform-*d*) δ 145.11, 144.54, 143.95, 142.09, 122.44, 110.55, 107.66, 104.60, 61.06, 56.57.

#### 3.1.10. 4,7-Dimethoxybenzofuran-5-carbaldehyde (**7**) via Halogen-Metal Exchange

*n*-BuLi (1.32 mL, 1.6 M in hexane, 2.1 equiv.) was added to a cooled (−78 °C) solution of compound **12** (0.26 g, 1.0 mmol, 1.0 equiv) in dry THF (5 mL) under argon atmosphere. The reaction mixture was stirred at −78 °C for 20 min. Then, DMF (0.25 mL, 3.25 equiv.) was added, and the cooling bath was removed. After 12 h, AcOEt (10 mL) was added, and the reaction was quenched with NH_4_Cl saturated aqueous solution. The organic layer was separated, washed with brine, dried over Na_2_SO_4_, and the solvent was evaporated to dryness. Chromatographic purification of the residue with A/P 1:4 *v*/*v* afforded the aldehyde **7** as a white crystalline solid. Melting point: 142–145 °C. Yield 50%. ^1^H NMR (400 MHz, Chloroform-*d*) δ 10.46 (s, 1H), 7.67 (dd, *J* = 2.3, 0.4 Hz, 1H), 7.25 (s, 1H), 7.02 (d, *J* = 2.3 Hz, 1H), 4.15 (s, 3H), 4.00 (s, 3H). ^13^C NMR (101 MHz, Chloroform-*d*) δ 189.09, 149.92, 145.30, 142.04, 122.24, 120.30, 105.88, 103.45, 61.74, 56.29.

#### 3.1.11. 1-Phenylprop-2-en-1-one (**13**) 

Formalin (0.8 mL, 37%, 10 mmol, 5 equiv.) was added to a solution of the ylide **8** (0.76 g, 2 mmol, 1 equiv.) in DCM (8 mL), and the mixture was heated at reflux overnight. After washing with brine, the organic phase was dried over anhydrous Na_2_SO_4_ and evaporated to dryness. Purification of the residue by column chromatography with A/P 1:4 *v*/*v* as the eluent phase furnished compound **13** as a clear oil. Yield 80%. ^1^H NMR (400 MHz, Chloroform-*d*) δ 7.97–7.93 (m, 2H), 7.58–7.55 (m, 1H), 7.51–7.45 (m, 2H), 7.16 (ddd, *J* = 17.2, 10.6, 0.6 Hz, 1H), 6.44 (dd, *J* = 17.2, 1.7 Hz, 1H), 5.95–5.91 (m, 1H). ^13^C NMR (101 MHz, Chloroform-*d*) δ 191.14, 137.35, 133.06, 132.47, 130.26, 129.00, 128.77, 128.70, 128.07 [45].

#### 3.1.12. 2-(2,5-Dimethoxyphenoxy) tetrahydro-2H-pyran (**14**) 

DHP (4.16 mL, 46 mmol, 10 equiv.) was added to a solution of compound **9** (0.71 g, 4.6 mmol, 1 equiv.) and PPTS (0.115 g, 0.46 mmol, 0.1 equiv.) in DCM (10 mL). The reaction mixture was stirred at room temperature overnight, then it was diluted with DCM and washed with NaOH solution and brine. The organic phase was dried over Na_2_SO_4_ and evaporated. Purification was performed by column chromatography using A/P 1:9 *v*/*v* as the eluent phase. Compound **14** was isolated as a clear oil. Yield 91%. ^1^H NMR (400 MHz, DMSO-*d*_6_) δ 6.89 (d, *J* = 8.8 Hz, 1H), 6.68 (d, *J* = 2.9 Hz, 1H), 6.50 (dd, *J* = 8.9, 2.9 Hz, 1H), 5.39 (t, *J* = 3.3 Hz, 1H), 3.82 (ddd, *J* = 11.1, 8.9, 3.8 Hz, 1H), 3.71 (s, 3H), 3.67 (s, 3H), 3.53 (dtd, *J* = 11.5, 4.3, 1.2 Hz, 1H), 1.94–1.38 (m, 7H) [44].

#### 3.1.13. 2-Hydroxy-3,6-dimethoxybenzaldehyde (**15**) 

*n*-BuLi (4 mL, 1.6 M in hexane, 6.37 mmol, 1.1 equiv.) was added dropwise to a cooled (0 °C) solution of compound **14** (1.38 g, 5.79 mmol, 1 equiv.) in anhydrous Et_2_O (60 mL) under argon atmosphere. The reaction mixture turned purple, then yellow, and stirring was continued at room temperature for 2 h. The reaction flask was cooled to −78 °C, and anhydrous DMF (1.78 mL, 23.16 mmol, 4 equiv.) was added. The reaction mixture was stirred at room temperature for 2 h, quenched with HCl (5 mL, 6 N), and stirred at room temperature for 1 h. The aqueous phase was extracted with AcOEt (×3), the combined organics were dried over Na_2_SO_4,_ and the solvent was evaporated. Purification of the residue was performed by column chromatography with A/P 1:4 *v*/*v* as the eluent phase. Compound 15 was isolated as bright yellow solid. Melting point: 67–69 °C. Yield 80%. ^1^H NMR (400 MHz, Chloroform-*d*) δ 12.17 (s, 1H), 10.31 (s, 1H), 7.02 (d, *J* = 8.9 Hz, 1H), 6.27 (d, *J* = 8.9 Hz, 1H), 3.84 (d, *J* = 0.8 Hz, 6H). ^13^C NMR (101 MHz, Chloroform-*d*) δ 194.91, 155.87, 153.58, 142.08, 120.27, 111.09, 99.36, 56.90, 55.78 [44].

#### 3.1.14. 3-Bromo-6-hydroxy-2,5-dimethoxybenzaldehyde (**16**)

AcONa (0.091 gr, 1.1 mmol, 1.1 equiv.) and then Br_2_ (0.056 mL, 1.1 mmol, 1.1 equiv.) was added to a cooled (0 °C) solution of compound **15** (0.182 g, 1 mmol, 1 eq) in acetic acid (3 mL). The reaction mixture was stirred at room temperature for 30 min. Then, H_2_O (10 mL) was added, and the mixture was extracted with DCM (10 mL × 3). The combined organic phases were washed with H_2_O (10 mL), dried over Na_2_SO_4_, and evaporated. Purification of the residue by column chromatography with A/P 1:4 *v*/*v* as the eluent phase furnished compound **16** as a light-yellow solid. Melting point: 92–95 °C. Yield 80%. ^1^H NMR (400 MHz, Chloroform-*d*) δ 11.89 (s, 1H), 10.21 (s, 1H), 7.19 (s, 1H), 3.92 (s, 3H), 3.87 (s, 3H). ^13^C NMR (101 MHz, Chloroform-*d*) δ 194.91, 152.63, 152.47, 145.51, 122.50, 115.15, 104.06, 63.41, 56.67 [44].

#### 3.1.15. Ethyl 2-(4-bromo-2-formyl-3,6-dimethoxyphenoxy)acetate (**17**)

Cs_2_CO_3_ (0.36 g, 1.1 mmol, 1.1 equiv.) and BrCH_2_COOEt (0.12 mL, 1.1 mmol, 1.1 equiv.) were added to a solution of compound **16** (0.261 g, 1 mmol, 1 equiv.) in DMF (5 mL). The reaction mixture was stirred at room temperature for 1 h. H_2_O (10 mL) was added, and the mixture was extracted with AcOEt (20 mL × 3). The combined organic phases were washed with H_2_O (20 mL), dried over Na_2_SO_4_, and the solvent was evaporated. Compound **17** was isolated as a white grainy solid after trituration with petroleum ether. Melting point: 85–86 °C. Yield 85%. ^1^H NMR (400 MHz, Chloroform-*d*) δ 10.49 (s, 1H), 7.27 (s, 1H), 4.76 (s, 2H), 4.23 (q, *J* = 7.1 Hz, 2H), 3.85 (d, *J* = 2.1 Hz, 6H), 1.28 (t, *J* = 7.1 Hz, 3H). ^13^C NMR (101 MHz, Chloroform-*d*) δ 189.32, 168.96, 150.74, 148.89, 148.75, 121.27, 112.82, 69.75, 62.64, 61.36, 56.66, 14.24.

#### 3.1.16. 5-Bromo-4,7-dimethoxybenzofuran (**12**) via Cyclization of (**17**)

LiOH·H_2_O (0.032 g, 0.78 mmol, 3 equiv.) was added to a magnetically stirred solution of compound **17** (0.09 g, 0.26 mmol, 1 equiv.) in a 3:1 THF/H_2_O (4 mL) mixture. After stirring at room temperature overnight, the reaction mixture was acidified with 1 M HCl and extracted with AcOEt (10 mL × 3). The combined organics were washed with brine, dried over anhydrous Na_2_SO_4,_ and evaporated to dryness. The white solid residue was dissolved in Ac_2_O (3 mL), AcONa (0.085 g, 1.035 mmol, 1.5 equiv.) was added, and the mixture was heated at 130 °C for 2 h. Successively, the temperature was lowered to 60 °C, and EtOH (3 mL) was added dropwise to consume acetic anhydride. The stirring was continued at 60 °C for 2 h. then, the reaction mixture was cooled to room temperature. H_2_O (3 mL) was added, and the mixture was extracted with AcOEt (10 mL). The organic phase was washed with NaOH (5 mL 2 N) and with H_2_O (5 mL), dried over Na_2_SO_4_, and evaporated. Chromatographic purification of the residue with A/P 0.5:9.5 provided compound 12 as a white crystalline solid. Melting point: 63–65 °C. Yield 57%.^1^H NMR (400 MHz, Chloroform-*d*) δ 7.59 (dd, *J* = 2.2, 0.5 Hz, 1H), 6.92 (s, 1H), 6.87 (dd, *J* = 2.1, 0.4 Hz, 1H), 3.97 (d, *J* = 1.7 Hz, 6H).^13^C NMR (101 MHz, Chloroform-*d*) δ 145.11, 144.54, 143.95, 142.09, 122.44, 110.55, 107.66, 104.60, 61.06, 56.57.

#### 3.1.17. Synthesis of (**1**) from (**12**) via Mizoroki-Heck

A reaction flask containing DMF (1 mL) and DIPEA (1 mL) was charged with phenylpropenone **13** (0.165 g, 1.25 mmol, 1.25 equiv.), aryl bromide **12** (0.257 g, 1 mmol, 1 equiv.), P(o-tol)_3_ (0.121 gr, 0.4 mmol), and Pd(OAc)_2_ (0.011 g, 0.05 mmol, 0.05 equiv.). The reaction mixture was stirred under an argon atmosphere at 110 °C for 5 h. Then, AcOEt (10 mL) and H_2_O (10 mL) were added to the residue after solvent evaporation. The insoluble solid was filtered on celite, and the organic phase was washed with brine and dried over Na_2_SO_4_. The solvent was evaporated, and Chromatographic purification of the residue with A/P 1:4 *v*/*v* provided compound **1** [10] as a yellow crystalline solid. Melting point: 98–103 °C. Yield 32%. ^1^H NMR (400 MHz, Chloroform-*d*) δ 8.22 (d, *J* = 15.8 Hz, 1H), 8.05–8.01 (m, 2H), 7.62 (d, *J* = 2.2 Hz, 1H), 7.59–7.48 (m, 4H), 7.05 (s, 1H), 6.95 (dd, J = 2.2, 0.6 Hz, 1H), 4.05 (s, 3H), 4.04 (s, 3H). ^13^C NMR (101 MHz, Chloroform-*d*) δ 191.00, 148.07, 144.97, 141.96, 140.32, 138.60, 132.53, 128.54, 128.50, 121.68, 121.47, 120.74, 105.34, 104.97, 61.51, 56.50. HPLC r.t. = 23,233 min. ESI = 309,2086 [M + H]^+^. HRMS *m/*z: [M + H]^+^ calc for C_19_H_16_O_4_ 309.11214, found 309.1121; [M + Na]^+^ calc 331.09408 found 331.0937. I.R. (neat) cm^−^^1^ = 3126, 1654, 1598, 1586, 1572, 1480, 1440, 1354, 1287, 1231, 1202, 1185, 1159, 1072, 1013, 995, 935, 875, 853, 771, 738, 718, 704, 685, 662.

#### 3.1.18. Synthesis of (**1**) from (**7**) via Wittig

A microwave vial was charged with aldehyde **7** (0.065 g, 0.3 mmol, 1 equiv.), ylide **8** (0.125 g, 0.33 mmol, 1.1 equiv.), and acetonitrile (3 mL); microwave was set at 135 °C for 1.5 h. The solvent was evaporated and the residue chromatographed by eluting with A/P 1:4 *v*/*v*. Yield 70%.

#### 3.1.19. (E)-3-(4,7-Dimethoxybenzofuran-2-yl)-1-phenylprop-2-en-1-one (**22**)

A microwave vial was charged with aldehyde (**20**, in mixture ratio 65:35 with 7) (0.08 g, 0.38 mmol, 1 equiv.), ylide **8** (0.144 g, 0.38 mmol, 1 equiv.), and acetonitrile (3 mL); microwave was set at 135 °C for 1.5 h. The solvent was evaporated and the residue chromatographed by eluting with A/P 1:4 *v*/*v*. Yield 70%. Melting Point: 155–160 °C. ^1^H NMR (400 MHz, Chloroform-*d*) δ 8.12–8.06 (m, 2H), 7.71 (d, *J* = 4.9 Hz, 2H), 7.59 (ddt, *J* = 8.3, 6.6, 1.4 Hz, 1H), 7.55–7.47 (m, 2H), 7.13 (s, 1H), 6.80 (d, *J* = 8.6 Hz, 1H), 6.54 (d, *J* = 8.6 Hz, 1H), 4.01 (s, 3H), 3.91 (s, 3H). ^13^C NMR (101 MHz, Chloroform-*d*) δ 189.04, 151.81, 147.38, 145.17, 139.38, 137.41, 132.41, 130.13, 128.09, 128.05, 121.06, 120.40, 110.05, 108.56, 102.49, 56.09, 55.25. HPLC r.t. = 22,617 min. ESI = 3,092,086 [M + 1]. HRMS *m/*z: [M + H]^+^ calc for C_19_H_16_O_4_ 309.11214, found 309.1122; [M + Na]^+^ calc 331.09408 found 331.094.

#### 3.1.20. Synthesis of (**22** and **1**) from (**20** and **7**) via Wittig by Convectional Heating

A round bottom flask vial was charged with aldehydes (**20**, in mixture ratio 65:35 with **7**) (0.08 g, 0.38 mmol, 1 equiv.), ylide 8 (0.144 g, 0.38 mmol, 1 equiv.), and toluene (3 mL); the mixture was refluxed overnight. After complete conversion of the starting material, the solvent was evaporated and the residue chromatographed by eluting with A/P 1:4 *v*/*v*. Yield 70%.

#### 3.1.21. (E)-3-(4,7-Dimethoxybenzofuran-6-yl)-1-phenylprop-2-en-1-one (**23**)

A microwave vial was charged with aldehyde **21** (0,08 mg, 0.38 mmol, 1 equiv.), ylide **8** (144 mg, 0.38 mmol, 1 equiv.), and acetonitrile (3 mL); the microwave was set at 135 °C for 1.5 h. The solvent was evaporated and the residue chromatographed by eluting with A/P 1:4 *v*/*v*. Yield 70%. Melting Point 138–141 °C ^1^H NMR (400 MHz, Chloroform-*d*) δ 8.23 (d, *J* = 15.9 Hz, 1H), 8.05–8.02 (m, 2H), 7.62 (d, *J* = 2.1 Hz, 1H), 7.61–7.49 (m, 4H), 6.89 (d, *J* = 2.1 Hz, 1H), 6.87 (s, 1H), 4.17 (s, 3H), 3.97 (s, 3H).^13^C NMR (101 MHz, Chloroform-*d*) δ 191.16, 148.59, 147.46, 145.45, 140.43, 138.72, 132.61, 128.64, 128.61, 122.92, 122.26, 122.11, 105.07, 101.56, 61.41, 55.92.HPLC r.t. = 24,183 min. ESI = 3,092,086 [M + 1]. HRMS *m/*z: [M + H]^+^ calc for C_19_H_16_O_4_ 309.11214, found 309.1121; [M + Na]^+^ calc 331.09408 found 331.0937.

#### 3.1.22. 2-(Dimethoxymethoxy)-1,4-dimethoxybenzene (**18**)

Bromoacetaldehyde dimethyl acetale (1.14 mL, 9.73 mmol, 1.5 equiv.) was added dropwise to a solution of **9** (1 g, 6.49 mmol, 1 equiv.), KOH (727 mg, 13 mmol, 2 equiv.) in DMAC (13 mL) at room temperature. The mixture was heated at 140 °C and stirred for 2 h until the starting material was no longer detected by TLC (A/P 1:4). It was allowed to cool down to room temperature and quenched with H_2_O (20 mL). The crude was extracted with Et_2_O (40 mL × 3), and combined organic phases were washed with an aqueous solution of 5% NaOH (30 mL) and H_2_O (30 mL). After drying over anhydrous Na_2_SO_4_, the ether extraction phase was evaporated in vacuo. The residue was purified by silica gel column chromatography (elution system A/P 1:4 *v*/*v*) to obtain **18** as a white crystalline solid. Yield 70%. ^1^H NMR (400 MHz, Chloroform-*d*) δ 6.80 (d, *J* = 8.8 Hz, 1H), 6.56 (d, *J* = 2.9 Hz, 1H), 6.44 (dd, *J* = 8.8, 2.8 Hz, 1H), 4.77 (t, *J* = 5.2 Hz, 1H), 4.03 (d, *J* = 5.2 Hz, 2H), 3.78 (d, *J* = 21.9 Hz, 6H), 3.46 (s, 6H). ^13^C NMR (101 MHz, Chloroform-*d*) δ 154.26, 149.08, 144.18, 113.21, 104.69, 102.73, 102.37, 69.16, 56.84, 55.73, 54.35.

#### 3.1.23. 4,7-Dimethoxybenzofuran (**19**)

Compound **18** (1 g, 4.12 mmol, 1 equiv.) and Sn-β zeolite (4.12 g) (kindly prepared by the analytical chemistry division of the University of Ferrara, Prof. Pasti Luisa) were poured in chlorobenzene (42 mL). The heterogeneous mixture was stirred at 105 °C. After 5 h, it was allowed to cool down to room temperature, filtered through a Gooch filter, and evaporated under reduced pressure. The residue was purified by silica gel column chromatography (elution system A/P 1:9 *v*/*v*) to obtain a white crystalline solid. Yield 62%. ^1^H NMR (400 MHz, Chloroform-*d*) δ 7.56 (d, *J* = 2.1 Hz, 1H), 6.86 (d, *J* = 2.1 Hz, 1H), 6.70 (d, *J* = 8.5 Hz, 1H), 6.54 (d, *J* = 8.5 Hz, 1H), 3.97 (s, 3H), 3.90 (s, 3H). ^13^C NMR (101 MHz, Chloroform-*d*) δ 147.60, 143.90, 140.38, 119.45, 106.43, 104.43, 102.72, 56.53, 55.80.

#### 3.1.24. 1-(2,5-Dimethoxyphenoxy)propan-2-one (**24**)

Compound **9** (560 mg, 4.21 mmol, 1 equiv.) was dissolved in DMF (0.2 M), then Cs_2_CO_3_ (1.5 g, 4.631 mmol, 1.1 equiv.) and bromoacetone (0.63 gr, 4.63 mmol, 1.1 equiv.) were added. The mixture was stirred overnight at room temperature and quenched with brine and H_2_O. The mixture was extracted with AcOEt, dried over Na_2_SO_4_, and the solvent was evaporated. The residue was purified by silica gel column chromatography (elution system A/P 1:4 *v*/*v*) to obtain 24 as a yellowish oil. Yield 63%.^1^H NMR (400 MHz, Chloroform-*d*) δ 6.83 (d, *J* = 8.8 Hz, 1H), 6.47 (dd, *J* = 8.8, 2.8 Hz, 1H), 6.39 (d, *J* = 2.8 Hz, 1H), 4.57 (s, 2H), 3.84 (s, 3H), 3.74 (s, 3H), 2.28 (s, 3H).^13^C NMR (101 MHz, Chloroform-*d*) δ 206.07, 154.22, 148.22, 144.00, 113.22, 105.29, 102.74, 74.46, 56.73, 55.77, 26.56.

#### 3.1.25. 4,7-Dimethoxy-3-methylbenzofuran (**25**)

A mixture of polyphosphoric acid (0.52 g) in chlorobenzene (15 mL) was heated at 120 °C, and a solution of 24 (370 mg, 1.55 mmol, 1 equiv.) in chlorobenzene (3 mL) was added. The mixture turned black, and after 20 min, it was allowed to cool at room temperature. The chlorobenzene phase was decanted, and the residue was rinsed three times with 20 mL of Et_2_O. The ether was evaporated, and cyclized product 25 was purified on silica gel column chromatography by eluting with A/P 0.2:9.8, *v*/*v*, to obtain a yellowish solid. Yield 50%. ^1^H NMR (400 MHz, Chloroform-*d*) δ 7.28 (d, *J* = 1.4 Hz, 1H), 6.66 (d, *J* = 8.5 Hz, 1H), 6.49 (d, *J* = 8.5 Hz, 1H), 3.95 (s, 3H), 3.86 (s, 3H), 2.35 (d, *J* = 1.3 Hz, 3H). ^13^C NMR (101 MHz, Chloroform-*d*) δ 149.34, 145.93, 140.50, 140.36, 120.11, 116.40, 106.33, 102.40, 56.58, 55.80, 29.78.

#### 3.1.26. (4,7-Dimethoxybenzofuran-3-yl)methanol (**26**), 4,7-Dimethoxybenzofuran-3-carbaldehyde (**27**)

Compound **25** (109 mg, 0.57 mmol, 1 equiv.) and SeO_2_ (125 mg, 1.134 mmol, 2 equiv.) in 1,4-dioxane were stirred at 75 °C for 96 h. The cooled mixture was filtered through Gooch, and the solvent evaporated under reduced pressure. Chromatographic purification of the residue using A/P 3:7 *v*/*v* as the eluent phase provided title compounds **26** and **27** in 1:1 ratio as brown and red solid, respectively. Yield 80%. ^1^H NMR (400 MHz, Chloroform-*d*) δ 7.47 (d, *J* = 1.0 Hz, 1H), 6.71 (d, *J* = 8.6 Hz, 1H), 6.56 (d, *J* = 8.6 Hz, 1H), 4.73 (d, *J* = 0.9 Hz, 2H), 3.95 (s, 3H), 3.94 (s, 3H), 2.85 (s, 1H). ^13^C NMR (101 MHz, Chloroform-*d*) δ 147.06, 146.03, 140.71, 140.55, 120.85, 118.54, 106.82, 102.70, 56.55, 55.94.

#### 3.1.27. 4,7-Dimethoxybenzofuran-3-carbaldehyde (**27**)

A solution of **26** (50 mg, 0,24 mmol, 1 equiv.) in anhydrous DCM (5 mL) was added to a suspension of PCC (129 mg, 0.6 mmol, 2.5 equiv.) in anhydrous DCM (5 mL). The red mixture turned dark brown and was stirred at room temperature overnight. The insoluble cake was rinsed three times with anhydrous Et_2_O. The ether portions were combined, filtered through a pad of Florisil, and evaporated. Compound **27** was afforded by chromatographic purification using A/P 3:7 *v*/*v* as the eluent phase. Yield 95%. ^1^H NMR (400 MHz, Chloroform-*d*) δ 10.41–10.41 (m, 1H), 8.21–8.18 (m, 1H), 6.77 (d, *J* = 8.6 Hz, 1H), 6.65 (d, *J* = 8.6 Hz, 1H), 3.96 (s, 3H), 3.92 (s, 3H). ^13^C NMR (101 MHz, Chloroform-*d*) δ 187.31, 148.52, 148.25, 145.99, 140.31, 123.46, 107.68, 104.36, 56.57, 55.82.

#### 3.1.28. (E)-3-(4,7-Dimethoxybenzofuran-3-yl)-1-phenylprop-2-en-1-one (**28**)

A microwave vial was charged with aldehyde **27** (65 mg, 0.3 mmol, 1 equiv.), ylide **8** (125 mg, 0.33 mmol, 1.1 equiv.), and acetonitrile (3 mL); the microwave was set at 135 °C for 2 h. The solvent was evaporated, and the residue was purified on silica gel chromatography eluting with A/P 1:4 v/v to obtain the desired product **28** as a yellow-ocher solid. Yield 65%. ^1^H NMR (400 MHz, Chloroform-*d*) δ 8.08–8.02 (m, 3H), 7.97–7.84 (m, 2H), 7.61–7.48 (m, 3H), 6.78 (d, *J* = 8.6 Hz, 1H), 6.65 (d, *J* = 8.7 Hz, 1H), 3.98 (s, 3H), 3.96 (s, 3H). ^13^C NMR (101 MHz, Chloroform-*d*) δ 190.58, 148.41, 146.56, 145.78, 140.38, 138.38, 134.75, 132.62, 128.56, 128.46, 124.00, 119.85, 116.75, 107.53, 104.04, 56.56, 55.96. HPLC r.t. = 23,017 min. ESI = 3,092,086 [M + 1]. HRMS *m/*z: [M + H]^+^ calc for C_19_H_16_O_4_ 309.11214, found 309.1121; [M + Na]^+^ calc 331.09408 found 331.0937.

#### 3.1.29. Cell Cultures and Stimulation

Bone marrow-derived macrophages (BMDMs) were isolated from C57BL/6 mice as described [52] and differentiated for 7 days in Iscove′s Modified Dulbecco′s Medium supplemented with 15% fetal bovine serum (FBS, Gibco), 1% penicillin/streptomycin (P/S) and 10 ng/mL M-CSF. THP-1 cells were grown in RPMI medium supplemented with 10% FBS, 100 U/mL penicillin, and 100 mg/mL streptomycin. THP-1 cells were stimulated by 100 ng/mL PMA overnight to differentiate into macrophages. All cells were grown in a 5% CO_2_ incubator at 37 °C. BMDMs were seeded at 5 × 10^5^ in 24 well plates. After 12 h, the medium was removed, and cells were treated with LPS from *Escherichia coli* 055:B5 (1 μg/mL) in fresh Iscove′s Modified Dulbecco′s Medium for 2 h. After that, the medium was removed and replaced with a serum-free medium containing DMSO or compounds (10 μM) for 30 min. Cells were then stimulated with Nigericin (10 μM) for 1 h. Human THP-1 cells were seeded at 3 × 10^5^ cells per well in 24 well plates. The following day, the overnight medium was replaced, and cells were stimulated with LPS (1 μg/mL) for 3 h. The medium was removed and replaced with a serum-free medium containing DMSO or compounds (10 μM) for 30 min. Cells were then stimulated with Nigericin (10 μM) for 1 h.

#### 3.1.30. In Vivo LPS Challenge

C57BL/6 mice were IP injected with compounds (25 mg/kg) or vehicle control (DMSO) 30 min before IP injection of LPS 1 mg/kg (4 h) and then were euthanized, and blood and peritoneal exudate were isolated. Mouse plasma was collected after blood centrifugation (1000× *g*, 15 min at 4 °C). ELISA for IL-1β was performed according to the manufacturer’s instructions (R&D Systems).

#### 3.1.31. ELISA

Supernatants from BMDMs and THP-1 cell culture were assayed for mouse or human IL-1β, respectively, by ELISA according to the manufacturer’s instructions (R&D Systems).

## 4. Conclusions

We succeeded in synthesizing the bioactive compound velutone F (**1**), the chalconoid contained in *Millettia velutina* stem, together with other 21 flavonoids. Our chemical total synthesis of velutone F is advantageous compare to the recently reported [24] seven steps semi-synthesis requiring the naturally occurring furanochromone derivative Khellin as the costly starting material (25 G > 800 USD). We could develop synthetic routes A and B; the first one establishes the benzofuran nucleus by creating the annellated carbocyclic ring onto a furan ring, while the second one proceeds exactly the other way. Instead, the Wittig olefination and the Heck coupling reaction are the featuring steps allowing for the assemblage of the 1,3-diaryl enone scaffold. Both the multi-step synthetic routes A and B establish the enone moiety at position C-5 of the 4,7-dimethoxybenzofuran thus, slightly modified synthetic approaches were designed in order to achieve the non-natural chalconoids **22**, **23**, and **28**, which formally are the C-2, C-3, and C-6 regioisomers of velutone F (**1**). The anti-inflammatory effects of the newly synthesized compounds are also reported.

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
