# Peer review of "Synthesis and NLRP3-Inflammasome Inhibitory Activity of the Naturally Occurring Velutone F and of Its Non-Natural Regioisomeric Chalconoids"

_ijms, 2022, doi:10.3390/ijms23168957_

Round 1

Reviewer 1 Report

The authors of the article take up the current topic of searching for analogues of a relatively new compound, which is velutone F. The search results in finding 3 active derivatives (22,23,28) with NLRP3-inflammasome inhibitors activity. The authors present ingenious pathways for the synthesis of individual substrates, demonstrating the knowledge of chemical synthesis. The obtained structures are confirmed by the spectra of NMR, IR, MS and HRMS spectroscopy. In addition, biological tests confirm very good, and sometimes better biological properties than the mentioned F velutone (1). Each section has a typical layout (Introduction, Results and discussion, Materials and methods, Conclusions). The whole thing is combined into a logically planned whole of research that has been properly carried out. The authors did not notice some shortcomings, which, however, do not affect the substantive reception of the work.

Main remarks:

1) Line 26: abbreviations should not be italicized.

2) Line 46: "Velutone F" should be in normal font, no italics.

3) If relationships 22, 23, 28 and others were new, it should be emphasized in the text, including the abstract.

4) Line 115: "Millettia velutina" should be in italics.

5) In the descriptions under the diagrams of all syntheses there is no differentiation of subscripts, eg it should be "H2SO4" and not "H2SO4", etc. Please correct.

6) "trans" in names should be italic.

7) Lines 146, 162: "O" should be in italics.

8) Line 221: why was step "d" not performed using microwaves? Please explain.

9) Entire article and charts: it should be "mL" and not "ml". The same for "ng/mL", "ug/mL" etc. Please correct.

10) Line 268: "in vitro" and "in vivo" should be in italics.

11) Line 235: there should probably be a different section number. Number 3 is the "Materials and Methods" section. Please check and correct the numbering of the individual sections.

12) The signature in Figure 3 should not be in italics.

13) Materials and methods: Please add manufacturers, cities and countries for each equipment used.

14) Line 293: on what equipment was the HRMS performed. Please add.

15) Section 3.18: it should be "g" and not "gr"

16) Line 527: "a microwave vial" - please add the type - glass, quartz, Teflon and its volume. If it is mentioned in some other synthesis, please also specify. This can also be added to the materials and methods at the beginning of section 3.

17) What was the purity of the chalcones used in biological research? On what basis was it determined? Please add.

18) Please write in the text of the manufacturer Sn-B-zeolite.

19) Line 581: it should be "Cs2CO3" and not "CsCO3".

20) Lines 610, 620: the compound name should be in italics.

21) Full article: there are sometimes missing spaces between numbers and units, numbers and signs of degrees Celsius. Please carefully track and correct.

22) I hope that the "non published materials" submitted for review will also be published as "supplementary materials"

Author Response

Ferrara, August 04 2022

Object: manuscript ID ijms-1839410, answer to Reviewer 1

Dear editor,

Thank you for having reviewed the manuscript and for considering it of potential interest for IJMS readers.

We report here below the reviewers’ questions / comments and our circumstantiated answers to all the points raised.

Reviewers answers:

Reviewer #1:

Comments:

R1        Line 26: abbreviations should not be italicized.

A1       Corrected

R2        Line 46: "Velutone F" should be in normal font, no italics.

A2        Corrected

R3        If relationships 22, 23, 28 and others were new, it should be emphasized in the text, including the abstract.

A3       The authors added in the text and also in the abstract a line to emphasized the novel molecular structures

R4        Line 115: "Millettia velutina" should be in italics.

A4       Corrected

R5        In the descriptions under the diagrams of all syntheses there is no differentiation of subscripts, eg it should be "H2SO4" and not "H2SO4", etc. Please correct.

A5       Corrected

R6        "trans" in names should be italic.

A6       Corrected

R7        Lines 146, 162: "O" should be in italics.

A7       Corrected

R8        Line 221: why was step "d" not performed using microwaves? Please explain.

A8       The reaction was carried out also in microwave condition, the yield and also the reaction time did not change from the classical heating and we decide to do the reaction in classical condition to increase the amount of compound synthesized.

R9        Entire article and charts: it should be "mL" and not "ml". The same for "ng/mL", "ug/mL" etc. Please correct.

A9       Corrected

R10      Line 268: "in vitro" and "in vivo" should be in italics.

A10     Corrected

R11      Line 235: there should probably be a different section number. Number 3 is the "Materials and Methods" section. Please check and correct the numbering of the individual sections.

A11     Corrected

R12      The signature in Figure 3 should not be in italics.

A12     Corrected

R13      Materials and methods: Please add manufacturers, cities and countries for each equipment used.

A13     We added in the material and methods all the information required by the reviewer.

R14      Line 293: on what equipment was the HRMS performed. Please add.

A14     We insert in the line 293 the instrument used for HRMS analysis

R15      Section 3.18: it should be "g" and not "gr"

A15     Corrected.

R16      Line 527: "a microwave vial" - please add the type - glass, quartz, Teflon and its volume. If it is mentioned in some other synthesis, please also specify. This can also be added to the materials and methods at the beginning of section 3.

A16     We added the information regarding the microwave vials in the materials and methods.

R17      ) What was the purity of the chalcones used in biological research? On what basis was it determined? Please add.

A17     The purity grade of the final chalcones used for biological analysis has been assessed by HPLC and HPLC-HRMS. All the HPLC spectra and HRMS spectra are in the supporting information file.

R18      Please write in the text of the manufacturer Sn-B-zeolite.

A18     done.

R19      Line 581: it should be "Cs2CO3" and not "CsCO3".

A19     Corrected

R20      Lines 610, 620: the compound name should be in italics.

A20     The authors changed the italics font.

R21      Full article: there are sometimes missing spaces between numbers and units, numbers and signs of degrees Celsius. Please carefully track and correct.

A21     Corrected

R22      I hope that the "non published materials" submitted for review will also be published as "supplementary materials"

A22     We hope so.

The new version of the manuscript should now fulfill the requirements for publication in the International Journal of Molecular Science.

       We look forward to hearing from you soon.

With my very best regards

                                                                             Prof. Claudio Trapella

Reviewer 2 Report

refer to the attached document 

Author Response

Ferrara, August 04 2022

Object: manuscript ID ijms-1839410, answer to Reviewer 2

Dear editor,

Thank you for having reviewed the manuscript and for considering it of potential interest for IJMS readers.

We report here below the reviewers’ questions / comments and our circumstantiated answers to all the points raised.

Reviewers answers:

Reviewer #2:

R1        line 8: 1Department of Chemistry change with 1Department of Chemistry

A1       Corrected

R2        line 78: in Figure 1 the word Velutone F it should be placed under the chemical structure and not sideways.

A2       Corrected

R3        scheme 1: remove the structure name “ 3-furanoic acid”, in a synthetic scheme it is not required to report the IUPAC nomenclature. In a similar way, remove “velutone F”, number 1 is sufficient.

A3       The IUPAC name has been removed from the schemes in all the manuscript.

R4        scheme 1: compound 4 has been synthesized starting from 3 (3.2. Methyl 3-(1,1,4-trimethoxy-4-oxobutyl)furan-2-carboxylate (4) [17]. A solution of compound (3) (0.9 gr, 3.75 mmol, 1 equiv.), HC(OMe)3(2.1 mL, 19.48 mmol, 5.2 equiv) line 319) while in the scheme it is synthesized starting from 2. Modify appropriately, in a similar way, compound 7 has been synthesized from 6.

A4       The scheme 1 has been changed to accomplish the referee request and has been clarified the reaction pathway.

R5        lines 119-123: Synthetic pathway A for target compound 1. Reagents and conditions: (a) i: LDA, THF, -78 °C, succinic anhydride; ii: dil.HCl; 35%; (b) MeOH, H2SO4, 90%; (c) HC(OMe)3, p-TsOH, MeOH rfx, 36 h, 90%; (d) i: t-BuOK, THF, -78 °C; ii: dil.HCl, 62%; (e) MeI, Cs2CO3, DMF, rt, 16 h, 96%; (f) i: LiAlH4, Et2O, 12 h; ii: PCC, CH2Cl2, rt, 1.5 h, 60% two steps; (g) MeCN, MW, 135 °C, 1.5 h, 70%. Use the appropriate formalisms. in particular the subscripts. Moreover, change rfx with reflux.

A5       We did the correction of all subscripts and also changed rfx with reflux.

R6        Line 130: tert-BuOH change with tert-BuOH.

A6       Corrected

R7        Scheme 2: as for the scheme 1 use the appropriate formalism (lines 136-143). Remove the IUPAC name (2,5-dimethoxy benzaldehyde) in the scheme.

A7       Corrected accordingly with scheme1 suggestion.

R8        Scheme 3: as the authors wrote “the resulting compound 18 was cyclized to 19 under the action of PPA (Sn zeolite also showed to efficiently promote this transformation [28]” but in the scheme 19 was synthesized from 8. Please make the concept clearer.

A8       The scheme was modified in accord with referee suggestion.

R9        lines 209-211: use the appropriate formalism and change rfx with reflux.

A9       Corrected

R10      scheme 4: as the authors wrote “compound 25 in two steps from phenol 8, namely: etherification with bromoacetone followed by acid-promoted cyclization of the resulting aryloxy acetone derivative 24” change the scheme appropriately.

A10     We did all the correction required by the reviewer.

R11      In the material and methods section starting from “A solution of compound (3) change with “A solution of compound 3” (lines 319-320) and in similar way “Title compound (4)” with “Title compound 4”. Please replace throughout the section (lines 330, 336, 340, 347, 351….

A11     All the correction required has been done.

The new version of the manuscript should now fulfill the requirements for publication in the International Journal of Molecular Science.

       We look forward to hearing from you soon.

With my very best regards

Prof. Claudio Trapella

Round 2

Reviewer 2 Report

in schemes 3 and 4 compound 8 (3,5 dimethoxy phenol as reported) correspond to compound 9 (see scheme 2). Compound 8 is 1-Phenyl-2-(triphenyl-5-phosphaneylidene) ethan-1-one (8) [16] as authors wrote. It must be added that the correct IUPAC name is 1-phenyl-2-(triphenylphosphoranylidene)ethanone. In the end if compound 8 was synthesized as reported in literature, it is not necessary to report the synthesis (see lines 384-394). Change the numbering in the text as well. 

Author Response

Dear reviewer,

Thank you for having reviewed the manuscript and for considering it of potential interest for IJMS readers.

We report here below the reviewers’ questions / comments and our circumstantiated answers to all the points raised.

Reviewers answers:

Reviewer #2:

R1: in schemes 3 and 4 compound 8 (3,5 dimethoxy phenol as reported) correspond to compound 9 (see scheme 2). Compound 8 is 1-Phenyl-2-(triphenyl-l5-phosphaneylidene) ethan-1-one (8) [16] as authors wrote. It must be added that the correct IUPAC name is 1-phenyl-2-(triphenylphosphoranylidene)ethanone. In the end if compound 8 was synthesized as reported in literature, it is not necessary to report the synthesis (see lines 384-394). Change the numbering in the text as well. 

A1: In schemes 3 and 4 we reported the right number of the compound, which it was 9 as the referee noticed.

At the line 94 the IUPAC name of 8 is been corrected.

Lines 391-410 is been removed as suggested from referee because the synthesis of 8 was reported in literature.

We changed numbers in the text as well as suggested from referee #2.

The new version of the manuscript should now fulfill the requirements for publication in the International Journal of Molecular Science.

We look forward to hearing from you soon.

With my very best regards

Prof. Claudio Trapella